# Inflammatory Microenvironment in Early Non-Small Cell Lung Cancer: Exploring the Predictive Value of Radiomics

**DOI:** 10.3390/cancers14143335

**Published:** 2022-07-08

**Authors:** Mariasole Perrone, Edoardo Raimondi, Matilde Costa, Gianluca Rasetto, Roberto Rizzati, Giovanni Lanza, Roberta Gafà, Giorgio Cavallesco, Nicola Tamburini, Pio Maniscalco, Maria Cristina Mantovani, Umberto Tebano, Manuela Coeli, Sonia Missiroli, Massimo Tilli, Paolo Pinton, Carlotta Giorgi, Francesco Fiorica

**Affiliations:** 1Department of Medical Sciences, Section of Experimental Medicine, Laboratory for Technologies of Advanced Therapies, University of Ferrara, 44121 Ferrara, Italy; prrmsl@unife.it (M.P.); mcristinamantovani@gmail.com (M.C.M.); sonia.missiroli@unife.it (S.M.); paolo.pinton@unife.it (P.P.); 2Radiology Division, SS.ma Annunziata Hospital, Azienda USL di Ferrara, 44121 Ferrara, Italy; e.raimondi@ausl.fe.it (E.R.); g.rasetto@ausl.fe.it (G.R.); r.rizzati@ausl.fe.it (R.R.); m.tilli@ausl.fe.it (M.T.); 3Tecnologie Avanzate TA Srl, Research & Development, 33100 Udine, Italy; matilde.costa@tecnologieavanzate.com; 4Department of Translational Medicine, University of Ferrara, 44121 Ferrara, Italy; giovanni.lanza@unife.it (G.L.); roberta.gafa@unife.it (R.G.); 5Department of Medical Sciences, Section of General and Thoracic Surgery, Sant’Anna University Hospital, 44121 Ferrara, Italy; giorgio.cavallesco@unife.it (G.C.); n.tamburini@ospfe.it (N.T.); p.maniscalco@ospfe.it (P.M.); 6Department of Radiation Oncology and Nuclear Medicine, AULSS 9 Scaligera, 37100 Verona, Italy; umberto.tebano@aulss9.veneto.it (U.T.); manuela.coeli@aulss9.veneto.it (M.C.)

**Keywords:** biomarkers, stage I NSCLC, radiomic features, microenvironment, inflammasomes

## Abstract

**Simple Summary:**

Despite therapeutic advances in treatments and translational research, patient prognosis for non-small cell lung cancer (NSCLC) remains unsatisfactory. Although stage I NSCLC has a relatively favorable prognosis than more advanced stage disease, almost 40% of patients diagnosed with a localized NSCLC are predicted to have a poor prognosis after surgery. The reasons why patients diagnosed with the same NSCLC traits will experience good or bad outcomes are far from being known. Therefore, new diagnostic paradigms are needed to predict treatment response and inform clinical decision-making. The tumor microenvironment is widely orchestrated by inflammatory cells and is an essential participant in carcinogenesis, contributing to the patient’s prognosis. This retrospective study explored the value use of computed tomography (CT) images as a “virtual biopsy”, in assessing the biomarkers CD68 and IL-1β that we have identified on a biological scale to be related to NSCLC patients’ overall survival (OS).

**Abstract:**

Patient prognosis is a critical consideration in the treatment decision-making process. Conventionally, patient outcome is related to tumor characteristics, the cancer spread, and the patients’ conditions. However, unexplained differences in survival time are often observed, even among patients with similar clinical and molecular tumor traits. This study investigated how inflammatory radiomic features can correlate with evidence-based biological analyses to provide translated value in assessing clinical outcomes in patients with NSCLC. We analyzed a group of 15 patients with stage I NSCLC who showed extremely different OS outcomes despite apparently harboring the same tumor characteristics. We thus analyzed the inflammatory levels in their tumor microenvironment (TME) either biologically or radiologically, focusing our attention on the NLRP3 cancer-dependent inflammasome pathway. We determined an NLRP3-dependent peritumoral inflammatory status correlated with the outcome of NSCLC patients, with markedly increased OS in those patients with a low rate of NLRP3 activation. We consistently extracted specific radiomic signatures that perfectly discriminated patients’ inflammatory levels and, therefore, their clinical outcomes. We developed and validated a radiomic model unleashing quantitative inflammatory features from CT images with an excellent performance to predict the evolution pattern of NSCLC tumors for a personalized and accelerated patient management in a non-invasive way.

## 1. Introduction

In the clinical management of cancer patients, assessments of prognosis are critical in the treatment decision-making process. Conventionally, patient outcome is related to tumor characteristics, the cancer spread, and the patients’ conditions. However, even when patients seem to have similar clinical and molecular features, substantial unexplained variation in patient survival time often remains. As suggested by William Bateson in 1908, it is very intriguing to “come up with an explanation“ about what makes the difference in these patients. Understanding the biological basis beyond these survival differences is very important in oncology for guiding the selection process of possible treatment choices for each patient. Furthermore, identifying these potential noninvasive biological features prioritizes optimizing therapeutic pathways.

We focused our attention on early NSCLC. Approximately 20–30% of NSCLC patients are diagnosed at earlier stages (stages I–IIIA), and patients with stage I disease especially have a relatively good prognosis [1]. The current standard treatment for early-stage NSCLC is surgery resection, though we do not have any clues on how to determine which patients may benefit significantly from other adjuvant therapies [2].

Three epidemiological studies [3,4,5] described and analyzed the survival results of stage I NSCLC patients, of whom more than 65% were alive after five years.

Similarly, we analyzed early NSCLC case studies at the University Hospital of Ferrara, and we identified patients with an unexpected short survival, defined as a duration shorter than the third interquartile survival (24 months), and patients with long survival, defined as a duration longer than the third interquartile survival (60 months).

We hypothesized that the reason for their OS differences was not hidden within the tumors since these patients developed tumors with the same traits but in the neighboring microenvironment. Indeed, the differences observed between patients affected by the same cancer type in terms of survival suggest that, in addition to the tumor itself, host factors strongly contribute to pathology progression. Accordingly, increasing evidence supports the idea that, during carcinogenesis, the crosstalk between cancer cells and host cells might be the key to discovering new diagnostic markers and developing more effective treatments [6].

In this scenario, the inflammatory levels within the TME emerged to have a fundamental role in sustaining carcinogenesis [7].

## 2. Materials and Methods

### 2.1. Patient Screening

This observational retrospective cohort study included 35 adults (≥18 years) newly diagnosed with stage I NSCLC (based on the 7th edition International Association for the Study of Lung Cancer tumor, node and metastasis classification of lung cancer) [8] at the University Hospital of Ferrara, Italy, between January 2009 and December 2010 and followed up until 2015. Routinely, all patients were assessed with the Eastern Cooperative Oncology Group (ECOG) performance-status score. The Charlson index [9] was retrospectively used to take comorbidities into account. Comorbidity scores were assigned without having any information regarding the outcome of the patients. From this patient database, we selected two groups of stage I NSCLC, in particular, a dataset of 15 patients in total, of which eight patients with long survivors and seven patients with short cancer survivors. All these 15 patients had a performance status of ≤2 and a life expectancy of more than three months, and they all received an active local treatment according to ESMO guidelines [10]. All patients in the study underwent a radical operation for lung cancer, and pathological examination verified tumor histology. None of the patients received radiotherapy or chemotherapy before surgery, and all patients were followed up after surgery. The informed consent was collected from patients whenever it was possible. The Local Ethics Committee supported this study (CE: 1016/2020/Oss/UniFe).

### 2.2. Immunohistochemistry in Tissue Specimens

Histological samples of the selected 15 patients were obtained from the archives, surgically resected, formalin-fixed, paraffin-embedded, and finally analyzed in the Department of Anatomy Pathology at the University Hospital of Ferrara, Italy. Histological series sections (4 µm thickness) of selected paraffin-embedded tumor specimens were prepared, and CD68 and IL-1β were evaluated by IHC staining according to previously reported methods [11]. Sections were then incubated O/N at room temperature with the antibodies against CD68 (PG-M1 Diagnostic Biosystems, Pleasanton, CA) and IL-1β (Abcam, Cambridge, UK). Two clinical pathologists independently assessed the IHC score. Staining was classified into five levels according to the percentage of positively stained cells per section as follows: 0 (absent), 0.5–5%; 1, 6–25%; 2, 26–50%; 3, 51–75%; 4, >75%. The staining intensity was deemed as follows: 0 (negative); 1 (weak); 2 (moderate); 3 (strong). Subsequently, the percentage and intensity scores were multiplied to achieve the total score (staining score = percentage score × intensity score).

### 2.3. Radiomic Feature Extraction on CT Images

A separate retrospective analysis was performed on the preoperative chest CT images of both groups of patients to explore the potential predictive value of radiomic features applied to NSCLC for OS.

The images analyzed in this study were acquired with different scanners and CT protocols: 3 patients were scanned with a GE CT scanner (HiSpeed, and Discovery models, GE Healthcare, Chicago, IL, USA), and 12 patients with a Philips CT scanner (Brilliance 64 and 256 slice models, Philips, Amsterdam, the Netherlands). Pixel size ranged from 0.49 × 0.49 mm^2^ to 0.98 × 0.98 mm^2^, with a slice thickness from 1.2 mm to 3 mm. The contrast-enhanced CT images in the portal phase with a standard reconstruction kernel were used for the radiomic analysis of all patients.

CT chest images acquired in the portal phase were processed by commercially available software version 11.7.174 (HealthMyne^®^, Madison, WI, USA) for NSCLC volume segmentation and radiomic features extraction.

A semi-automated segmentation algorithm was adopted to capture the entire lesion volume for each lung lesion. The expert drew an initial axis crossing longitudinally the lesion on an axial plane; then, the algorithm automatically computed the volume segmentation [12,13,14,15]. We aimed to explore the concept of “virtual biopsy” by drawing sampling spheres with a fixed diameter of 1 cm to obtain a focused sample for radiomic feature extraction, inclusive of both tumor and peritumoral components. The sampling sphere eliminates the differences in the sampled volumes among the CT images. Two spheres were centered on the automatic tumor contour for each lesion and placed respectively with the central slice on axial and coronal planes. An additional sampling sphere was placed within the muscle tissue, drawn on the same slice of the axial sphere, to assess the image noise’s influence on the radiomic features (Figure 1). From each of the outlined volumes of interest (VOIs), a vast number of radiomic features were obtained, including local intensity, intensity-based statistical and intensity histogram features, texture, and high order statistical features. The feature calculation was by the IBSI indications regarding the implementation and the feature aggregation methods [16]. The whole segmentation process was then repeated by a second radiologist, achieving a double set of VOIs, in order to evaluate the robustness and reproducibility of the quantitative data analysis. A cross-evaluation of the extracted features from both data sets was applied to find the most significant features.

### 2.4. Statistical Analysis

Data from both patient groups were analyzed using MedCalc statistical software version 14.8.1 (MedCalc Software Ltd., Ostend, Belgium).

Patients’ characteristics and initial treatment were reported with descriptive statistics. The median time from diagnosis to first treatment received was calculated for treated patients. The Kaplan–Meier method [17] estimated OS from the date of diagnosis to death.

For immunohistochemistry analysis, the averages of staining scores of two groups were compared. The statistical method included Student’s unpaired *t*-test; *p*-values are reported in the figure legend.

All radiomics quantitative parameters are reported as median value and interquartile range (IQR). The Mann–Whitney U test assessed differences between NSCLC-derived radiomics metrics measured in patients with short or long OS extracted from preoperative CT images. A separate analysis was performed for each set of VOIs. A *p*-value ≤ 0.05 was considered statistically significant. Only metrics showing a *p*-value equal to or less than 0.05 on both analyses were considered. To assess the diagnostic performance of the radiomic metrics, a receiver operating characteristic (ROC) analysis was performed. The cut-off values were calculated from the Youden Index.

## 3. Results

### 3.1. Clinical Outcomes

Between January 2009 and December 2011, 240 patients were diagnosed with NSCLC at Ferrara University Hospital and treated. Approximately 30% were diagnosed with localized disease (stages I–IIIA), of whom 35 (48.6%) were identified as having stage I disease according to the 7th edition American Joint Committee on Cancer (AJCC) cancer staging guidelines [8] (Appendix A). The median age of stage I NSCLC patients was 74 years, and 27 of these patients were male (77.1%). Table 1 reports the characteristics of all analyzed patients, 57.1% were adenocarcinoma, and 42.9% were squamous cell carcinoma.

Among all patients, the median Charlson comorbidity index was 4 (range 2–8). All patients were treated radically with only localized treatment, including surgery (91.4%) and stereotactic radiotherapy (8.6%). For all 35 patients, 5-year OS was 59.7%, and the survival of the third interquartile was 24 months (Figure 2A). Within these selected patients, we identified a group with long overall survival (>than the third interquartile survival time) and a group with poor survival (Figure 2B).

These patients had the same tumor characteristics (stage I and G1 or G2), same Eastern Cooperative Oncology Group (ECOG) performance score (0 and 1) and same Charlson comorbidity index (3 and 4), establishing that poor survival was not related to comorbidities, performance status, cancer staging and/or cancer grading. In the long survival group (8 patients), the 5-year OS rate was 100%; in contrast, in the poor survival group (7 patients), no patients were alive at 5 years (*p* < 0.0001).

### 3.2. Biological Outcomes

The tumor microenvironment (TME), where macrophages are a fundamental component [18], performs multiple functions and significantly modulates clinical outcome [19]. Considering the difference in survival between the two groups of patients, we investigated whether there were differences at the histological level in the TME in terms of inflammatory levels.

We used the CD68 marker to stain the amount of tumor-associated macrophages and the IL-1β as a signal for the NLRP3 inflammasone activation.

Immunohistochemical analysis of human lung cancer specimens obtained from the two different groups of patients revealed that high CD68 expression levels in the peritumoral space correlated with strong positivity for IL-1β in patients with a poor prognosis (Figure 3). In contrast, immunohistochemical analysis of patients with long survival showed low positivity for both CD68 and IL-1β, suggesting that a pro-inflammatory TME correlates with a poor prognosis in early NSCLC.

### 3.3. Radiomic Outcomes

Among the different features analyzed from each set of measurements, a total of six metrics showed a statistically significant difference in value distribution between the two groups of patients in both data sets, namely, four first-order statistical features (intensity-based energy, maximum intensity, median intensity, and 90th intensity percentile) and two texture features (autocorrelation measured on the gray level co-occurrence matrix and computed from each 3D directional matrix and averaged over the 3D directions, and gray level non-uniformity measured on the gray level run length matrix and computed from each 2D directional matrix and averaged over 2D directions and slices) (Figure 1); the same features extracted from muscle tissue VOIs showed no significant differences between the two patient groups. Detailed median values, IQRs and 95% confidence interval (CI) of each feature in both sets of data are reported in Figure 4.

The ROC curve reference values for each feature and each data set, along with their respective sensitivity and specificity percentages, are shown in Figure 5.

## 4. Discussion

Prognosis data on stage I NSCLC are very limited, despite a homogenous pattern driving treatment approach and generally reflecting ESMO guidelines: surgery and radiotherapy [10]. In this study, the estimated OS for patients diagnosed with stage I disease was 94.1% at one year after diagnosis and 59.7% at five years after diagnosis with 33 and 21 patients alive, respectively. Our data are comparable to data from three published previous studies [3,4,5]. We focused our attention on a substantial but unexplained variation in survival among stage I NSCLC patients ranging from 0 to 100%.

To shed light on these results, we have tried to select outlier patients with similar individual characteristics (performance status, comorbidities, age), same disease stage (Ia and Ib), same tumor grade, and comparable histology results. Our initial hypothesis was that differences in the biological profiles of the environment neighboring tumors could explain the extremely different OS outcomes for those patients. Understanding these biological determinants in extreme outlier cases could reveal useful biological features and a route for improving individualized patient responses [20]. After identifying patients with similar tumor characteristics, the survival difference was based on other factors. Thus, we moved away from a cancer-centric vision; the objective was to focus on the cells forming the tumor stroma [21] and their interaction with the tumor.

It is well established that the TME and, above all, the inflammatory response of the host, represent critical determinants of cancer cell growth and proliferation [22]. In this context, inflammasomes, multiprotein complexes of the innate immune system, regulate the release of proinflammatory cytokines such as interleukin-1β and interleukin-18 [23]. Dysregulation of these inflammasomes could lead to favorable conditions for tumor growth. Among the most studied inflammatory processes, the NLRP3 inflammasome is the best characterized, and it has attracted attention for its involvement in several cancer types, although its role is still being debated and controversial. Macrophages are the main actors in the immune NLRP3 inflammasome-mediated response and trigger signaling cascades that lead to the synthesis and secretion of IL-1β. This macrophage-produced cytokine is a potent chemotactic factor that allows for cell migration to sites of injury and inflamed tissues. In addition, IL-1β secreted into the TME is mainly proinflammatory and promotes tumorigenesis, tumor invasiveness, and immunosuppression [24].

CD68 is a well-known glycosylated glycoprotein that is highly expressed in macrophages and other mononuclear phagocytes [25]. For this reason, CD68 alone or in combination with other cell markers of tumor-associated macrophages (TAMs) is a good predictive value of survival in cancer patients.

Accordingly, we evaluated CD68 and IL-1β scores in histological lung cancer specimens from early NSCLC patients with poor survival and long survival. We retrospectively investigated the biological characteristics of the lung TME in extreme outlier patients with stage I NSCLC and evaluated the role of NLRP3-mediated inflammation.

In the analysis, the histological lung cancer specimens from non-surviving patients displayed a distinct proinflammatory profile characterized by significantly higher levels of CD68 (macrophages) and IL-1β scores than the samples isolated from surviving patients.

Therefore, our data suggest that CD68 and IL-1β could be promising biomarkers for lung tumor staging and prognosis assessment in the clinical diagnosis process.

Subsequently, we evaluated if this biological difference could be reflected in noninvasive imaging. All patients had preoperative CT images used for staging. We hypothesized that CT-derived indicators may change when stromal reorganization, inflammatory cell infiltration, and endothelial activation occur in the tumor site. In our patients, we histopathologically confirmed these microenvironmental changes, and we tested our hypothesis with CT. We used a so-called “virtual biopsy” in which the CT scans were analyzed by a high-powered computing method called radiomics. We applied this method to extract additional information from the data-rich images acquired by CT, especially from the region between the tumor and healthy tissue. A sampling sphere was placed between the tumor and healthy tissue to analyze healthy and pathological tissue simultaneously, leading to an improved evaluation of CT characteristics. In the analysis described above, we found six promising radiomic features that characterized the two groups of patients, and these differences matched not only the histopathological differences but also the observed differences in OS.

It appears that the proven difference in the microenvironment could be recognizable on routine CT scans of the lung when radiomic sampling is applied. Among those six radiomic features, two were texture features, and four were intensity features (therefore directly tied to the density of the tissues found in the microenvironment) [26]. Karimi et al. identified increased density on CT as a marker to detect and quantify early subclinical pathological changes [27]. Additionally, a positive correlation was demonstrated between lung density and measures of local inflammation, including both total cell concentration and concentrations of macrophages and neutrophils in bronchoalveolar lavage fluid.

In a preclinical model, researchers have demonstrated that there is a strong correlation between quantitative measurements, such as lung density, total lung volume, and volume of the air spaces or lung tissues, and gold-standard histological measurements [28]. In particular, Marien et al. revealed that quantitative measurements from micro-CT showed longitudinal changes in the lungs in metastatic mouse model [29]. Furthermore, histopathological studies have revealed that primary tumors in the premetastatic phase demonstrate different important characteristics, such as immunosuppression, inflammation, angiogenesis/vascular permeability, lymphangiogenesis, organotropism, and reprogramming [28]. Sun et al. analyzed both the microenvironment of solid tumors and radiomic features to evaluate CD8+ cells in the tumor periphery and found a promising way to predict immune phenotypes and clinical outcomes for solid cancer patients [30].

In our study, the patients with more attenuated tumor margins on CT showed a shorter OS than the patients with less dense tumor margins. Patients with longer OS showed higher values for the two texture features. Thus, we hypothesized that a proinflammatory microenvironment (high levels of CD68 and IL-1β) has pronounced chemotactic activity that elicits more inflammatory infiltrate. This increased cellularity results in greater density in the CT scan. Considering that there is a close association between the higher proinflammatory profile of the microenvironment and shorter OS, it appears plausible that these differences could lead to changes in the radiomic features and be recognized noninvasively through CT scans.

With a standardized feature extraction process, the advantage of radiomics includes its noninvasiveness, rapid processing times, and applicability/repeatability since the method is based on the analysis of diagnostic CT images, which are always available.

Nonetheless, our study has some limitations. As a retrospective study, the CT scans were acquired with different CT machines; furthermore, the number of participants was low, which makes the subgroup analysis results uncertain. Thereby, to overcome these limitations, the structural differences between the machines were nullified by the periodical calibration that every machine receives. Relative to the differences in the acquisition parameters, the concept of “virtual biopsy” (including cancer and healthy tissue) allowed us to obtain a more homogeneous dataset. Moreover, along with the abovementioned spheres, for every patient, a third sphere was drawn, located on the muscle tissue. This third sphere was used to reduce any possible bias created by noise: when a radiomic feature differed between the two different muscle spheres, it was considered influenced by noise and therefore excluded from the virtual biopsy. Finally, for every patient, two sets of spheres were drawn, doubling the overall size of the dataset.

## 5. Conclusions

The strength of our study is that we studied only the extremes of the survival curve of patients, bridging laboratory and clinical studies. Histopathological changes that take place in the lung microenvironment are the basis that connect laboratories and clinics. We demonstrated the comparability of radiomic features extracted from CT images with evidence that emerged from the IHC study validating the results of both analyses. Our findings provide evidence for investigating NSCLC at an early stage by analyzing radiomic information obtained from CT images. We have already started the selection of a more extensive case series to find other extreme cases and corroborate this evidence in a different kind of microenvironment in the two groups of patients and infer an underlying explanation for this difference. Furthermore, a prospective study is being planned to demonstrate the predictive value of this combination of radiomic and biological features.

The analysis of extreme outlier cases allowed us to reveal unique biological and radiomic characteristics in a patient setting; for this reason, applying this information to a more heterogeneous case series will be the focus of future work. If confirmed by further investigations, our hypothesis could lead to a better understanding of tumoral behavior, facilitating universally accepted tailored treatment for patients.

## Figures and Tables

**Figure 1 cancers-14-03335-f001:**
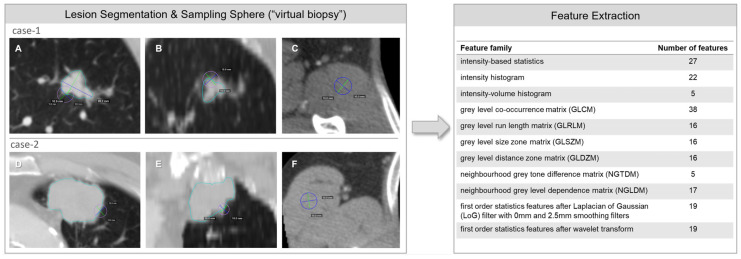
Description of radiomic features extraction. On the left, examples of volumes of interest delineated on the CT images: the light blue line represents the contour of the lesion volume ((**A**,**D**) on the axial plane, (**B**,**E**) on the coronal plane), the purple line represents the sampling spheres (“virtual biopsy”) centered on the boundary between tumor and peritumoral components ((**A**,**D**) on the axial plane, (**B**,**E**) on the coronal plane), and the blue line represents the sampling sphere within the muscle tissue (**C**,**F**). The two rows represent two different patients. The table on the right shows the list of families and the number of the extracted radiomic features for each patient.

**Figure 2 cancers-14-03335-f002:**
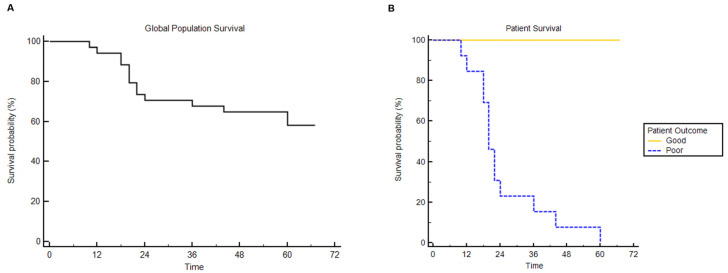
Overall survival analysis. Overall survival curve for all 35 patients evaluated (**A**); (**B**) overall survival for 15 patients defined as long and short survival and with the same characteristic for ECOG, comorbidity and tumor staging, histology, and grading.

**Figure 3 cancers-14-03335-f003:**
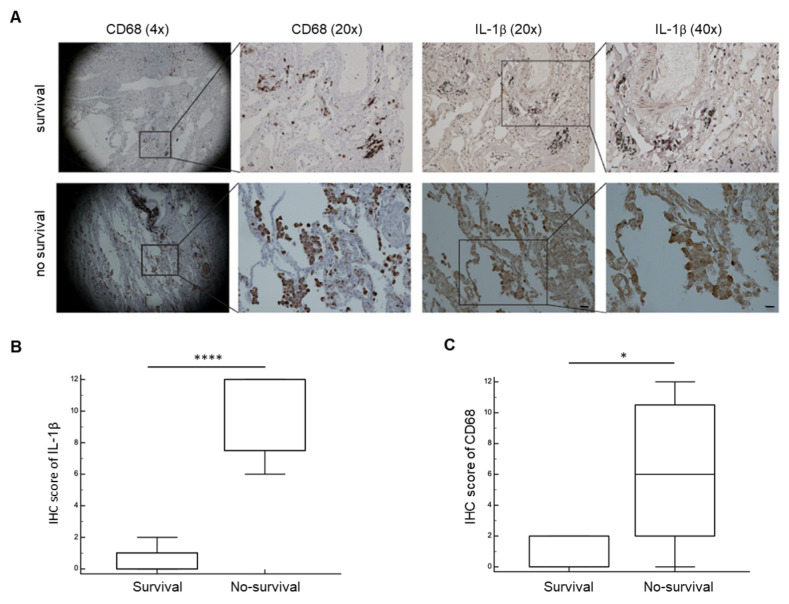
Biological analysis of tumor microenvironment in lung tissue. (**A**) Representative immunohistochemical images for CD68 (macrophages) and IL-1β expression in lung tumor tissue and adjacent normal tissues. The boxed areas are enlarged to show staining for CD68 and IL-1β. Magnification 4×, 20×, and 40×; (**B**) analysis of IHC scores of IL-1β expression in lung tumor tissues and adjacent normal tissues (*n* = 14); (**C**) analysis of IHC scores of CD68 expression in lung tumor tissues and adjacent normal tissues (*n* = 14). *p*–values were determined by an unpaired *t*-test. Bars: S.E.M. * *p* < 0.05; **** *p* < 0.0001.

**Figure 4 cancers-14-03335-f004:**
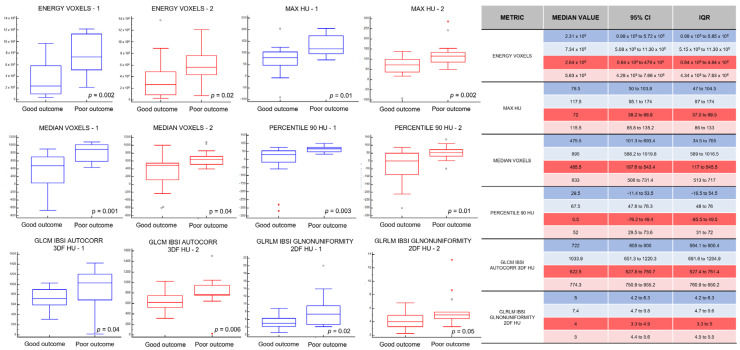
Description of selected features. On the left, the box plots for each selected feature. In the box plot, the central box represents the values from the lower to upper quartile (25 to 75 percentile). The middle line represents the median. The vertical line extends from the minimum to the maximum value, excluding outside and far out values which are displayed as separate points (respectively, marked as black circles and red triangles). The blue box plots represent the results obtained with the segmentation and feature extraction performed by the first radiologist. The red box plots display the results obtained by a second radiologist. On the right, the median value of the box plots, the 95% (CI), and the (IQR) values for each relevant feature are reported in the table. The results are differentiated by different colors: blue and red colors indicate, respectively, the first and the second radiologists, while, for each group, the first row represents patients with a long survival (good outcome) and the second row represents patients with short survival (poor outcome). Feature acronyms: ENERGY VOXELS = IBSI-consistent intensity-based energy that represents a measure of the magnitude of raw voxel values, MAX HU = IBSI-consistent maximum of the Hounsfield Unit values, MEDIAN VOXELS = IBSI-consistent median of the raw voxel values, PERCENTILE 90 HU = IBSI-consistent lowest intensity that is greater than 90% of the voxel intensities (expressed in Hounsfield Unit), GLCM IBSI AUTOCORR 3DF HU = IBSI-consistent Autocorrelation measured on the Gray Level Co-occurrence Matrix and computed from each 3D directional matrix and averaged over the 3D directions (expressed in Hounsfield Unit), GLRLM IBSI GLNONUNIFORMITY 2DF HU = IBSI-consistent Gray level non-uniformity measured on the Gray Level Run Length Matrix and computed from each 2D directional matrix and averaged over 2D directions and slices (expressed in Hounsfield Unit).

**Figure 5 cancers-14-03335-f005:**
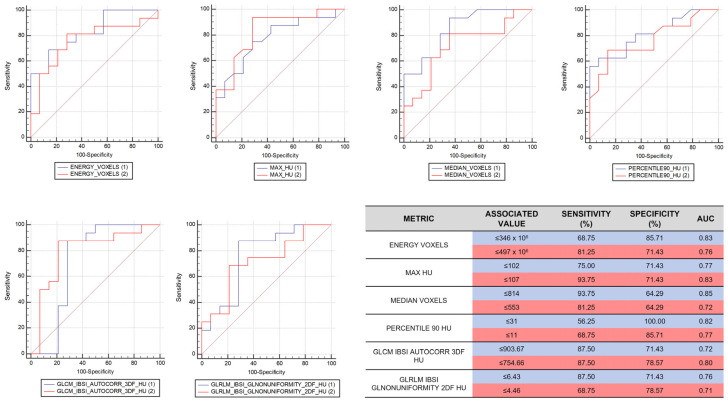
The (ROC) curves for each selected feature. In the table on the right, the values, the sensitivity, the specificity, and the Area Under Curve (AUC) of each relevant feature are reported. In both ROC curves and rows of the table, the blue color represents the results obtained by the first radiologist, while the red color is related to the second radiologist.

**Table 1 cancers-14-03335-t001:** Descriptive features of patients.

	All Patients Stage I (*n* = 35)	Long Survival Group (*n* = 8)	Short Survival Group (*n* = 7)
age (mean, range)	72 (42–84)	72.4 (66–80)	73.6 (65–78)
male/female	27/8	4/4	2/5
ECOG 0	20 (57.1%)	8 (100%)	7 (100%)
ECOG 1	14 (40%)	-	-
ECOG 2	1 (2.9%)	-	-
Charlson index (mean, range)	4 (2–8)		
adenocarcinoma	20 (57.1%)	5 (62.5%)	5 (71.4%)
squamous cell carcinoma	15 (42.9%)	3 (37.5%)	2 (28.6%)
grading G1	7 (20%)	1 (12.5%)	2 (28.6%)
grading G2	22 (62.8%)	6 (75%)	4 (57.1%)
grading G3	6 (17.2%)	1 (12.5%)	1 (14.3%)
EGFR mutations	12 (34.3%)	4 (50%)	4 (57.1%)
Stage Ia	20 (57.1%)	4 (50%)	5 (71.4%)
Stage Ib	15 (42.9%)	4 (50%)	2 (28.6%)
Surgery/Stereotactic Radiotherapy	32/3	8/0	7/0

## Data Availability

The anonymized images used in the current work may be available on request to the corresponding authors.

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
