# Peer review of "Inflammatory Microenvironment in Early Non-Small Cell Lung Cancer: Exploring the Predictive Value of Radiomics"

_cancers, 2022, doi:10.3390/cancers14143335_

Round 1
Reviewer 1 Report
I thank the authors for this very interesting work. It reports the place of radiomic in the management of bronchial cancers, a very promising tool in the therapeutic decision.
However, I have a few comments.
The authors do not report the mutation status of the patients which is a prognostic factor (EGFR, KRAS, ALK...).
Why the authors were only interested in macrophages and IHC CD68 and IL-1β expression, tumor infiltration by TREGs also seems to be an important prognostic factor. Why not evaluate it.
Is there a significant difference between the good outcome and poor outcome groups on radiomic results? Statistical data is missing, p value in figure 4.
It seems important to clarify the correlation between IHC and radiomics.
Author Response
Reviewer 1
I thank the authors for this very interesting work. It reports the place of radiomic in the management of bronchial cancers, a very promising tool in the therapeutic decision. However, I have a few comments. Our response: We are indebted to Reviewer #1 to appreciate our research article. We have done our best to further improve the manuscript and figures following the constructive suggestions of the reviewers.
The authors do not report the mutation status of the patients which is a prognostic factor (EGFR, KRAS, ALK...). Our response: We thank Reviewer #1 for his suggestions and consideration of our work. However, we would like to point out that since this is a retrospective research work that examines samples collected between 2009 and 2011, a period in which the required mutation markers had not been taken into account for the development of the research project itself. We were able to find out the data on EGFR mutations that we have added in the table. Nevertheless, patients were selected with similar histopathological characteristics to have homogeneous groups. We would certainly like to develop this aspect as we continue with this new diagnostic method.
Why the authors were only interested in macrophages and IHC CD68 and IL-1β expression, tumor infiltration by TREGs also seems to be an important prognostic factor. Why not evaluate it. Our response: We fully agree with Reviewer #1 that Tregs are important prognostic factors. However, this is a retrospective research work that examines samples collected between 2009 and 2011, and with the material we had available, we could not investigate other markers. Moreover, IHC staining only provides information on the amount of infiltrating Tregs but not on their functionality. We would certainly like to develop this aspect as we continue with a new prospective study.
Is there a significant difference between the good outcome and poor outcome groups on radiomic results? Statistical data is missing, the p-value in figure 4. Our response: We have amended this issue in the figure set. We thank Reviewer #1 for spotting it out.
It seems important to clarify the correlation between IHC and radiomics. Our response: We thank Reviewer #1 for the suggestion. IHC was the only biological analysis that could be carried out on retrospective samples. We used IHC staining to validate new radiomic features we have extracted from CT images. The novelty of this study lies in having identified new inflammatory features that reflect the inflammatory state of the tumor microenvironment and correlate with the patients’ outcomes. We have more fully explained this correlation in the conclusions of the manuscript and in the abstract graphic.
Reviewer 2 Report
A significant contribution to radionomics in cancer of that should be published for further peer discussion. The literature has been carefully collated and assessed with a clear and justified conclusion, and methodologicals well developed and applied.
Addresses an important problem in a systematic way. This should impact on the field and will be of value to those working in the area.
Have the authors considered the longitudinal study?
There are some minor spelling and grammatical errors to be tidied up, and some of the references require a second checking, but otherwise a strong piece. I endorse publication.
Author Response
Reviewer 2
A significant contribution to radiomics in cancer of that should be published for further peer discussion. The literature has been carefully collated and assessed with a clear and justified conclusion, and methodological well developed and applied. Addresses an important problem in a systematic way. This should impact the field and will be of value to those working in the area.
Our response: We thank Reviewer #2 for the enthusiastic evaluation of our work. We have done our best to address the constructive comments from Reviewers to ameliorate the quality of our manuscript further.
Have the authors considered the longitudinal study?
Our response: We thank Reviewer #2 for the suggestion. However, this is a proof-of-concept clinical trial to understand if there is a correlation between patients’ outcomes and biological characteristics of the tumor microenvironment and also with radiomic features. We are carrying out a longitudinal study with a higher number of patients enrolled which allows us to strengthen the evidence of this preliminary study.
There are some minor spelling and grammatical errors to be tidied up, and some of the references require a second checking, but otherwise a strong piece. I endorse publication.
Our response: We have checked for English grammar and corrected errors as requested.
Round 2
Reviewer 1 Report
Thanks to the authors